# Cytokine Modulation in Breast Cancer Patients Undergoing Radiotherapy: A Revision of the Most Recent Studies

**DOI:** 10.3390/ijms20020382

**Published:** 2019-01-17

**Authors:** Raffaella Marconi, Annalisa Serafini, Anna Giovanetti, Cecilia Bartoleschi, Maria Chiara Pardini, Gianluca Bossi, Lidia Strigari

**Affiliations:** 1Laboratory of Medical Physics and Expert Systems, IRCCS Regina Elena National Cancer Institute, 00144 Rome, Italy; annalisa.serafini13@gmail.com (A.S.); gianluca.bossi@ifo.gov.it (G.B.); 2Laboratory of Biosafety and Risk Assessment, Division of Health Technologies, Department of Sustainable Territorial and Production Systems, ENEA, 00123 Rome, Italy; anna.giovanetti@enea.it (A.G.); cecilia.bartoleschi@enea.it (C.B.); mariachiara.pardini@enea.it (M.C.P.)

**Keywords:** cytokines, breast cancer, radiotherapy, biomarkers

## Abstract

Breast cancer (BC) is the most common tumor and the second cause for cancer-related death in women worldwide, although combined treatments are well-established interventions. Several effects seem to be responsible for poor outcomes in advanced or triple-negative BC patients. Focusing on the interaction of ionizing radiation with tumor and normal tissues, the role of cytokine modulation as a surrogate of immunomodulation must still be explored. In this work, we carried out an overview of studies published in the last five years involving the cytokine profile in BC patients undergoing radiotherapy. The goal of this review was to evaluate the profile and modulation of major cytokines and interleukins as potential biomarkers of survival, treatment response, and toxicity in BC patient undergoing radiotherapy. Out of 47 retrieved papers selected using PubMed search, 15 fulfilled the inclusion criteria. Different studies reported that the modulation of specific cytokines was time- and treatment-dependent. Radiotherapy (RT) induces the modulation of inflammatory cytokines up to 6 months for most of the analyzed cytokines, which in some cases can persist up to several years post-treatment. The role of specific cytokines as prognostic and predictive of radiotherapy outcome is critically discussed.

## 1. Introduction

Breast cancer (BC) is the most common tumor and the second cause for cancer-related death in women worldwide [1]. The therapeutic possibilities are surgery, radiation therapy/radiotherapy (RT), chemotherapy (CHT), hormone (endocrine) therapy (HT), and targeted therapy [2], or a combination of these can increase therapeutic efficacy. Several effects seem to be responsible for poor outcomes in advanced or triple-negative BC patients. 

Enhancing the radiation dose delivered to the tumor and reducing the toxic effect in normal tissue and critical organs nearby are the main RT goals [3]. Clinically it is well known that cellular response to radiation is influenced by many factors such as total dose, fractionation, tumor replication time, hypoxia, and tumor radiosensitivity [4]. Thus, the choice of total radiation dose, number of fractions, and overall treatment time is critical for improving therapeutic outcome. Currently, thanks to the performance of new technologies, the dose can be finely modulated allowing for dose-per-fraction increases (hypofractionation). This approach has also been applied to BC patients resulting in lower healthcare costs [5]. 

Of note, the main mechanism causing cell death due to radiation is DNA strand breaking, but numerous recent evidence studies suggest the involvement of non-DNA-related mechanisms [6]. Moreover, preclinical and clinical evidence has shown RT to induce antitumor immune responses mediating the regression of non-irradiated tumors or metastases distant from the irradiation site, a process known as the abscopal effect [7]. The indirect anticancer effect of RT has been clinically observed since 1952 in different human malignancies in areas separated from the irradiated tissue [8]. In 2003, Camphausen et al. [9] reported TP53 as an important mediator in eliciting the abscopal effect and identified a radiation-dose dependency to induce this effect. Results obtained by our research group, in accordance with this study, revealed p53 status and delivered dose as strictly required to trigger the abscopal effect [10]. The abscopal effect has been sporadically reported in the literature during clinical practice and is still not exploited for treatment purposes [11], although involvement of the immune system is widely accepted by the scientific community. 

Tumors have evolved multiple ways to evade the immune system by creating a strong immunosuppressive microenvironment [12,13]. Cancer is also able to exert a dysregulation of the physiological cytokine milieu, ultimately tipping the balance between immunosuppression and immunostimulation, thus influencing disease progression [14]. Ionizing radiation is able to induce an inflammatory response involving cells of the innate immune system, especially macrophages [15] leading to chronic inflammation through tissue damage and fibrosis [16], eliciting immunogenic cell death via the upregulation or release of danger-associated molecular patterns (DAMPs). The release of DAMPs is dose-dependent and has been shown to recruit and activate dendritic cells, myeloid-derived suppressor cells, and effector T cells by secretion of different cytokines [17]. 

Accordingly, the understanding of the immune cell context in the tumor microenvironment is crucial in predicting the response to a given therapy and improving clinical outcomes [18]. In this context, cytokines can affect both innate and adaptive immune responses [19] and can be secreted by a broad range of immune cells, prompting pro- or anti-inflammatory responses based on their combination [20]. In the tumor microenvironment, cytokines interact with a plethora of biomolecules, such as cancer stem cells, microRNA, epithelial-mesenchymal transition markers, and transcription factors, and are involved in processes such as epigenetic regulation, autophagy, immunoediting, and inflammation related to tumor progression [21]. These features prompted the design of various cytokine-based approaches in cancer immunotherapy to achieve more efficiency and efficacy in treatments. 

Cancer and RT can significantly alter overall cytokine secretion: similar to a molecular fingerprint, the “cytokine profile” or “signature” can be a molecular tool used to identify not only different cellular subsets [22], but also new drug candidates [23], and it is currently under investigation as a tumor biomarker [24]. 

Moreover, RT can influence early and late radiation side effects, which are very similar to chronic inflammatory disease, paradoxically beneficial for the tumor [25]. 

Ultimately, BC represents a perfect model where immunotherapeutic strategies can be studied because it is a very common disease, presents many possibilities to design clinical trials, and has a strong immunophenotypic variety [26]. In this review, we focused on overviewing the most relevant literature regarding cytokines, BC patients, and RT in the last five years. Our main aim was to evaluate whether RT can influence cytokine modulations and if one or more relevant cytokine profiles significantly correlate with treatment outcomes. Particular attention was focused on results obtained from serum analysis, since this represents a favorable approach to biomarker identification/validation for disease diagnosis and/or prognosis. The possibility of assessing robust serum biomarkers is in fact an essential prerequisite for cancer treatment optimization and personalization.

## 2. Results

For this review, 45 studies were retrieved from PubMed and another 2 by searching the bibliographies of other papers. Of these, only 15 studies [27,28,29,30,31,32,33,34,35,36,37,38,39,40,41] were included in the analysis, and the reasons for exclusion are reported and detailed in the Appendix A. Results of the studies included in our analysis are reported in Table 1 and Table 2. 

Of the selected studies, 67% reported cytokine concentration values calculated on blood/serum samples of BC patients, 27% reported values evaluated on tissue samples (biopsy or surgery specimen), and one study reported calculated TGF-β1 concentrations quantified by ELISA on the supernatant from wound fluid (WF) draining from the surgical operation sites of BC patients [34].

Two studies [42,43] did not include detailed or recoverable information regarding RT schedule for patient treatment, and are reported in a separate table (Table 3). 

We decided to also evaluate these two studies because they reported cytokine concentration values calculated before the beginning of post-surgery treatments, thus representing “basal” values for the focus of our research (identifying which cytokines modulate upon treatment and correlate with patient outcomes). Only a few studies generally report basal values for these potential disease biomarkers. Of the selected studies, in fact, only 6 out of the 17 studies included basal values of cytokines analyzed on blood/serum [27,29,31,35,41,43].

The number of patients enrolled in the studies ranged from 23 to 534, and results from the analyzed studies are presented in the following paragraphs.

### 2.1. Interleukins (IL-2, IL-6, IL-8, and IL-10)

Most of the studies reported data on IL-2, IL-6, IL-8, and IL-10 as the most significantly modulated cytokines in BC patients receiving radiotherapy. 

IL-2 shows an increase in serum level in BC patients compared to healthy donors at baseline and 1-month post-RT; this difference is lost after 6 months [39]. IL-2 is the most prominent candidate as a diagnostic biomarker in tumors.

Significant differences in IL-6 serum levels between healthy donors and BC patients at early stage are also reported [29,39], showing also a progressive trend related to tumor stage [31]. 

Specifically, the study by Shibayama et al. measuring IL-6 plasma levels in 105 BC surgical patients within 1 year after adjuvant RT reported higher cytokine levels in RT-treated BC patients with respect to unexposed patients [40], and the elevation of plasma IL-6 levels partially explains the relationship between therapy and the cognitive impairment in treated BC patients. Similarly, De Sanctis et al. [39] reported increased IL-6 serum levels in BC patients one month after RT, with no differences registered at 6 months post-RT. In accordance with this, Schmidt et al. showed increased IL-6 serum levels at 3 weeks from the beginning of RT but not at 6 weeks after RT [35]. These findings indicate that IL-6 serum level peaks are a highly time-dependent early biomarker, as corroborated by Westbury et al. [38] which showed no significant differences in IL-6 serum levels in BC patients at 2 or 5 years post-RT as assessed during the monitoring of RT-induced fibrosis. 

Overall, studies revealed that the modulation of IL-6 after RT depends on the timepoints selected for the investigation. It is noteworthy that the increased IL-6 serum level after multiple sessions of RT, reported in many studies, has often been linked to worse therapeutic outcome or toxicity. However, the use of the IL-6 serum level as a prognostic and predictive biomarker of successful treatment is still an open question.

Indeed, serum IL-8, along with IL-6, was reported as significantly decreased in RT responder BC patients when compared to the baseline levels [31]. In contrast, De Sanctis et al. [39] registered no variations in IL-8 serum level at 4 weeks after RT when compared to baseline, whereas Muraro et al. [29] reported a lower IL-8 level in BC patients compared to controls at baseline and a significant increase 1 month after stereotactic body RT (SBRT). Thus, the use of the IL-8 serum level as a prognostic and predictive biomarker is still not conclusive.

In addition, the significance of the anti-inflammatory cytokine IL-10 in BC is controversial. IL-10 serum levels were reported as not differently modulated in BC patients and healthy controls [29], neither at baseline nor one month after RT [39]. 

Of relevance, the possibility of using a combined panel of cytokines may represent a step forward in the estimation/prediction of the therapeutic outcome/toxicity. The serum levels of IL-6, IL-8, and IL-10 evaluated before RT have been reported as potential biomarkers of metastasis in BC patients, with their levels being higher than in healthy controls [31]. In particular, regarding a prognostic role, the IL-6, IL-8, and IL-10 serum levels were significantly reduced in good RT responders compared to the baseline levels, as well as in BC patients with mild conditions compared to patients with severe conditions. 

### 2.2. Other Cytokines

Regarding TNF-α, its role is still debated, together with that of other inflammatory cytokines such as IL-8 or transcriptional factors (i.e., FOXP3+). The TNF-α serum level in BC patients is higher when compared to healthy controls as reported in De Sanctis et al. [39], whereas it showed no variation in Wang and Yang [31]. 

One month after RT, TNFα is unchanged in Muraro et al. [29] and Wang and Yang [31]; whereas TNFα is reported as the only cytokine significantly modulated with respect to the basal level even six months after RT [39].

In Wang and Yang [31], authors also indicate SCC-Ag and CYFRA 21-1 serum level as potential biomarkers of metastasis and prognosis for BC patients treated with RT, with significant differences reported in the survival of the SCC-Ag negative group compared to that of the SCC-Ag positive group. Indeed, a median survival of 25 and 16 months was reported for the SCC-Ag negative and positive groups, respectively. 

### 2.3. Transforming Growth Factor-β-1 (TGF-β1) 

Ciftci et al. [43] reported a higher level of the pro-inflammatory and pro-fibrotic cytokine TGF-β1 associated with better survival after two years’ follow-up in metastatic BC patients treated with CHT or combined CHT and RT, although no detailed information regarding the RT schedule for patients was specified.

Interestingly, Boothe et al. [41], analyzing TGF-β1 serum levels during intravenous accelerated hypofractionated partial breast irradiation (APBI), highlighted the relevance of the timing for TGF-β1 dosage, since they reported a progressive lowering of the cytokine serum level also one month after treatment. Moreover, higher levels of TGF-β1 were also found in wound fluid drained 24 h after surgery in early stage BC patients [34]. The cytokine has shown to be biologically active in inhibiting the proliferation of primary lymphatic endothelial cells suggesting TGF-β1 as a prognostic biomarker in both serum and wound fluid for fibrosis development.

In contrast to Boothe et al. [41], Scherer et al. [34] reported high TGF-β1 serum levels after RT, but given the different site of measurements (wound fluid vs serum), time after RT (24 h vs. 1 month), and type of RT treatment (intraoperative radiotherapy (IORT) vs. APBI), it is difficult to speculate which factor can determine the different outcome, although timing seems to play a crucial role. Of relevance, effects of TGF-β1 can be seen also six months or even years after RT: high protein levels in skin biopsies lead to abrogated lymphatic vessel formation, associated with microvascular damage [37].

### 2.4. Interferon Gamma (IFN-γ)

The IFN-γ serum levels in BC patients are not different from those revealed in healthy donors, even one month after RT [39], making questionable the role of IFN-γ as tumor or prognostic biomarker.

Todorović-Raković et al. [30] adopted longer timepoints analyzing the protein and mRNA at the intratumoral level up to 14 years after RT and revealed a consistent prognostic association of IFN-γ protein with a poor disease outcome in patients with a longer follow-up (>7 years). Similarly, for IFN-γ mRNA, a peculiar time-dependent prognostic shift has been reported: from 6 months after RT and until 7 years, IFN-γ mRNA is associated with a good outcome, whereas in patients with a longer follow-up (>7 years), it has been related to a poor prognosis. 

### 2.5. Chemokines

Strom et al. [32] developed two gene-expression signatures (GESs) to define radiosensitivity index (RSI) and immune-activation, the latter by selecting 12-Chemokines (12-CK) (CCL2, CCL3, CCL4, CCL5, CCL8, CCL18, CCL19, CCL21, CXCL9, CXCL10, CXCL11, and CXCL13). The prognostic value of the RSI and the 12-CK GESs were explored in an independent dataset of 282 BC patients treated with surgery and RT, and showed RSI-low status (associated with more radiosensitive tumors) and 12-CK-high immune-active status independently associated with improved distant metastasis-free survival (MFS). However, the study does not report details on the radiation dose used and the sampling time after RT. Tudoran et al. [36] analyzed the gene expression of 84 inflammatory molecules and their receptors, including the chemokines CXCL13 and CCL26 in 40 peripheral blood samples from patients with Her2-primary BC to study the association of triple-negative phenotype with age, clinical stage, tumor size, lymph nodes, and menopausal status. However, no information regarding the RT treatment schedule (i.e., the total dose and fractionation) was included in this study. 

## 3. Discussion

Cancer progression and treatment response can be influenced by interactions between tumor and host cells such as fibroblasts, vessels, white blood cells, and various soluble molecules [44]. In our work, we evaluated recently published clinical studies reporting cytokine profiles of BC patients before and after RT. Breast cancer heterogeneity makes it challenging to diagnose and treat the solid tumor [45], and the availability of specific biomarkers associated with outcome helps personalize and optimize treatment. 

In general, in the case of some interleukins (Figure 1), evidence of their potential correlation with treatment outcome and/or toxicity is emerging.

Regarding the reported studies, five of them show a variation of cytokine levels after RT in BC patients [27,31,35,39,41]. From our analysis, we identified IL-2, IL-6, and TGFβ as the cytokines most significantly modulated by the tumor itself and by RT. 

IL-6 and IL-2 levels appear to correlate with the BC patient outcome and could be considered in combination as putative tumor biomarkers as well as a prognostic marker for RT-induced toxicity (i.e., erythema and fatigue) [39,46]. 

IL-2 serum levels are higher in BC patients compared to healthy individuals. Similar findings are also reported in colorectal [47], endometrial [48], and pancreatic cancer [49]. 

Of note, IL-2 is the first cytokine approved as an effective immunotherapy in melanoma and renal cancer, due to its ability to expand T cells [50]. Recchia et al. [51] reported that a complex therapeutic regimen combining hormone therapy, CHT, RT, and immunotherapy had a positive effect on 20-year survival in younger metastatic BC patients. This study also indicated that IL-2 and 13-cis retinoic acid administration was able to restore immune function, pointing out a possible therapeutic use for this cytokine. Moreover, a phase I clinical trial where IL-2 is given in combination with RT and L19, a protein targeting the extra domain B of fibronectin, is ongoing for patients with oligometastatic solid tumors [52].

IL-6 peak is associated with the worst prognosis because it is related both to tumor stage and time to RT response. As reported in Wang and Yang [31], IL-6 serum levels after treatments can be used as an indicator to understand which BC patients need additional treatments, thereby going towards personalized therapies, although further studies are needed to support this result. This study reported a significant reduction in the IL-6 serum level in RT responders compared to non-responders, but the cohort of BC patients was heterogeneous for tumor stages and treatment types, including also combined CHT-RT (CRT). Interestingly, physical exercise has shown to be beneficial in counteracting fatigue and pain by inhibiting IL-6 peak [35], therefore finding a way to reduce IL-6 peak appears to be a promising strategy for new therapeutic approaches for BC.

Corroborating the observations in BC patients, IL-6 serum levels are higher compared to healthy controls also in lung [53], bladder [54], ovarian [55], head and neck [56], and prostate cancer [57], and in all cases the IL-6 serum level is associated with worse prognosis. Moreover, increased IL-6 serum levels correlate with increased pulmonary toxicity for lung cancer after RT [58]. Positive IL-6 staining in tissue biopsies of esophageal carcinoma were significantly associated with shorter survival, especially after RT without surgery [59].

Regarding inflammatory cytokines, IL-8 and TNFα findings are contradictory. Besides the studies previously cited in this review, TNFα serum levels are higher only in patients with advanced stage BC [60]. Moreover, other studies, although they consider only BC patients treated with CHT, associate high levels of serum TNF-α with a reduced risk of cancer progression [61].

The serum level of IL-8 is reported to increase [60,62] compared to healthy individuals. However, measurements performed after RT present contradictory results making their use as diagnostic and prognostic markers questionable at present. 

A similar variability of results is also found for the anti-inflammatory cytokine IL-10 for BC patients, although in other cancers, according to a meta-analysis of 21 studies [63], this IL has been reported to be associated with worse outcome. In addition, and in contrast to IL-6, IL-10 is not modulated in BC patients by physical exercise [64]. 

Increased levels of IL-6 in the serum and tumor site have been demonstrated in several cancers including BC [65]. While this increase is usually accompanied with poor prognosis and lower survival in BC patients, downregulation of IL-6 is related to better response to treatment [28,66]. 

Interestingly, polymorphisms of the IL-6R gene can also affect the BC prognosis [67]. These studies suggest that genetic variants of IL-6 and IL-6R can help diagnosis and better treatment or prevention, but we still need further investigations to highlight their relation.

Inhibitors of IL-6 or the selective targeting of the IL-6–sIL-6R complex resulted in novel and more effective therapeutic strategies for BC treatment [68]. The Janus kinase (JAK) inhibitors could inhibit tumor growth in gastric and colorectal cancer via suppressing the IL-6/JAK2/signal transducer and activator of transcription 3 (STAT3) pathway and enhancing the secretion of anti-tumor cytokines. Little is known about the efficacy of JAK inhibitors for BC treatment, and further investigations are needed.

Other cytokines appear to be more specific and modulated in BC patients. For example, transcriptional factor FOXP3 expression correlates with better overall survival (OS). On the contrary, specifically related to hormone receptor negative BC, a higher infiltration of FOXP3+ Tregs in the stromal site was associated with poor prognosis in patients treated with anti-Her2 therapy, although no significant difference was found when it came to the intratumoral site [42]. However, these findings were obtained on bioptic specimens only, and no evidence is currently available in serum analysis.

Notwithstanding contradictory literature [69,70,71,72], the two most recent studies [41,43] agreed that higher TGF-β1 serum levels are present in BC patients before any treatment when compared to healthy controls. However, at present, the prognostic value of the TGF-β1 serum level is still difficult to assess. In Boothe et al. [41], high TGF-β1 serum levels correlate significantly with increased RT-induced fibrosis suggesting its prognostic value in assessing a patient’s tolerance to radiation dose escalation. In Ciftci et al. [43], OS improvement is correlated with higher TGF-β1 serum levels. Even though the two studies adopted different endpoints, the results associating elevated TGF-β1serum levels with both positive and negative effects appear controversial. This can be explained by the fact that in Boothe et al. [41] all patients were at stage 0–1 and non-metastatic, whereas in Ciftci et al. [43], the positive correlation of the TGF-β1 serum level to survival was observed only in metastatic BC patients. 

Tumor stage, site/time of measurements, and choice of RT treatment appear to be crucial in order to use TGF-β1 as a prognostic marker for BC. It appears that high serum levels of TGF-β1 correlate with better survival, but there are worse RT-related side effects. More exhaustive analyses are required to better understand the possible TGF-β1 prognostic values.

IFN-γ cannot be considered a tumor biomarker, although it seemed to be a promising tool for cancer therapies: unfortunately, results obtained in different cancer types turned out to be contradictory [73]. Very preliminary results for the potential use of IFN-γ as a therapeutic agent to counteract cutaneous radiation syndrome after RT were previously reported to be effective for a cohort of five patients including two BC patients [74]. At this time, clinical trials are still ongoing for IFN-γ as a treatment for Her-2 positive BC, as well as ovarian, fallopian tube, and primary peritoneal cancer, and soft tissue sarcoma [75].

Some studies also reported survival curves estimating different endpoints (i.e., progression-free survival (PFS), MFS, or OS) in correlation to individual cytokine level modulation [29,30,32,42,43], thus indicating possible biomarkers useful for patient stratification. For example, a low basal level of IL-10 before SBRT correlates to a high probability of PFS, as well as a high NF-κB level in natural killer cells [29]. These preliminary findings need to be validated in larger patient cohorts. 

Of note, most of the selected studies included in our work analyzed cytokine serum level because the work was less invasive, less expensive, and easier to perform, although we also considered works reporting cytokine levels in tissue biopsies and draining fluids, not only for protein level but also for mRNA. 

Regarding methodological approaches adopted to assess cytokine variation, some methods, such as GES, are not currently applicable to the routine assessment of treatment outcomes for BC patients due to the very high costs and level of expertise required for performing such high-throughput analyses. None of the analyzed studies, moreover, reported specific information regarding the costs that cytokine profiling in diagnosis/prognosis assessment would add to national healthcare systems. The latter aspect has to be taken into consideration in modern healthcare management and budget allocation.

## 4. Materials and Methods

### Study Search Strategy and Selection Criteria

A PubMed search was performed using the query string reported below to review only studies involving the evaluation, modulation, and assessment of interleukins and cytokines in BC patients treated with RT.

Query: ((Interleukin OR cytokine) AND “breast cancer” AND radiotherapy) NOT (“in vitro” OR cell line OR mouse) NOT laser.

The research was restricted to the last five years in order to include only the most recently published studies. The search was done on 12 November 2018.

PRISMA methodology was used for selecting studies based on the following criteria. Titles and abstracts were independently reviewed by two authors in order to decide study inclusion. In the case of controversial judgment, the paper was evaluated by a third author. Full articles were retrieved when the abstract was considered relevant and only papers published in English were considered. The bibliographies of retrieved papers and reviews were also examined to identify other relevant articles to be included. Papers were considered eligible when reporting results of cytokine profiles evaluated in BC patients that included RT as a treatment option. Studies where BC patients underwent immunotherapy have been excluded in order to avoid potential confounding factors.

The PRISMA flowchart (see Figure 2) summarizes the searching strategy adopted in this study.

Excluded studies are listed in the Appendix A with the reason of their exclusion. 

## 5. Conclusions

It is now well accepted that cancer is able to modulate cytokine levels to promote tumor progression, and many cytokines are altered in cancer patients compared to healthy controls, as also reported in this review. Recent evidence suggests that cancer therapies also have an influence on the cytokine milieu: this modulation can contribute to their efficacy. In this review, we chose to focus on cytokine levels in BC patients and how the cytokines could be altered by RT, in order to evaluate their reliability as tumor biomarkers or prognostic markers, not only for overall survival but also with respect to collateral effects. Given that external beam RT technologies have evolved dramatically over the past few decades, and due to the astonishing technical improvements achieved in imaging techniques and hypofractionation, it is currently possible to select the most opportune approach in order to optimize BC patient treatment.

## Figures and Tables

**Figure 1 ijms-20-00382-f001:**
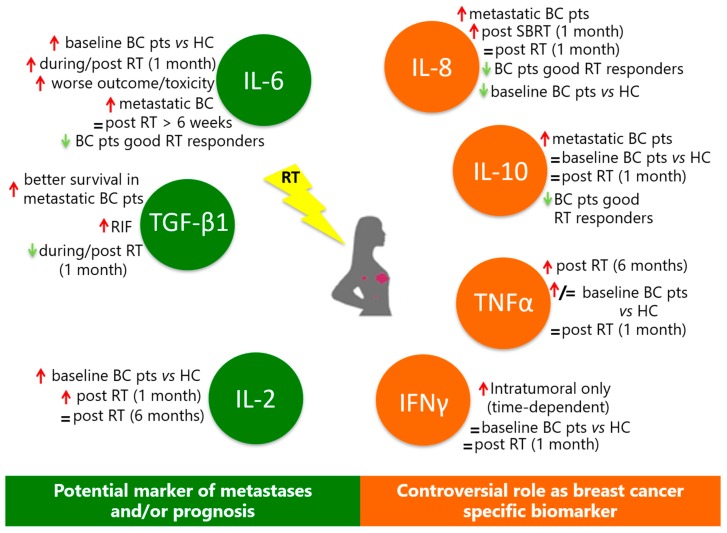
Summary of main examples of evidence regarding cytokine modulation in BC patients undergoing radiotherapy from an analysis of clinical studies published in the last five years. Red and green arrows indicate increase or decrease of cytokine concentrations, respectively.

**Figure 2 ijms-20-00382-f002:**
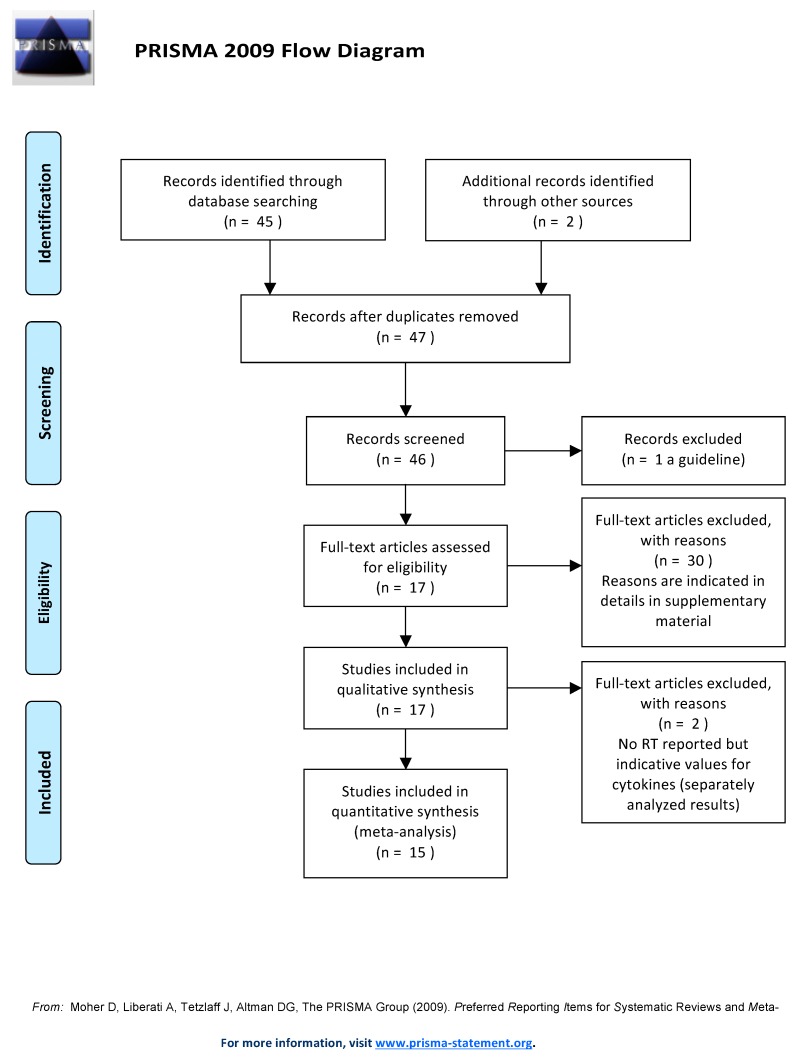
PRISMA flowchart reporting the searching strategy adopted in this study.

**Table 1 ijms-20-00382-t001:** Type of treatment and number of enrolled patients/healthy controls in the studies published in the last five years regarding cytokine profiling in BC patients undergoing radiotherapy (RT). See Table 2 for additional details.

Ref	No. of BC Patients and CTR	Treatment (S, RT, CHT, CRT Combined)	RT TD and d in frs	CHT/HT
[27]	73	Adj RT	50 Gy and 2 Gy or 42.56 Gy and 2.66 Gy	No
[28]	209 BC + 106 CTR	S, RT, and CHT	Not mentioned	Not mentioned
[29]	21 oligometastatic BC + 14 healthy donors	SBRT	30 Gy and 10 Gy	Concomitant CHT/HT and trastuzumab
[30]	73	S + RT	Not mentioned	No
[31]	534 BC (treated) and 452 CTR	RT (66.7%); RT + CHT (33.3%)	50–74 Gy and 1.8–2.0 Gy	1 to 6 cycles of platinum-based CHT
[32]	282	S and post-operative RT	40–74 Gy and 1.8–2 Gy	No adjuvant systemic therapy (i.e., CHT or HT)
[33]	149	Post S treatment including additional CHT, HT, or RT was determined by the individual treating physician	In 128 pts (50 Gy and 2 Gy)	4 cycles of anthracycline based neo-adj CHT with or without taxane (given before or after the anthracycline) followed by S. Trastuzumab to patients with Her-2 amplified disease
[34]	11 (treated) and 12 CTR	IORT	20 Gy and 20 Gy	No
[35]	103	Adj RT	50 Gy and 2 Gy	55% of patients received HT
[36]	40	Not mentioned	Not mentioned	Steroids, CHT, or S
[37]	40 (treated) and 8 CTR	CHT yes/no: 30/18	50 Gy (45–76) and 2–2.5 Gy	Bisphosphonate treatment
[38]	26 cases and 44 CTR	S and RT	50 Gy in 25 frs; 41 Gy in 13 frs or 39 Gy in 13 frs, boost 10 Gy in 4 frs	Tamoxifen yes/no: 61/9 Adj; CHT yes/no: 22/48
[39]	40 (treated) and 10 healthy donors	RT	50 Gy in 25 frs; 50 Gy in 25 frs + boost of 10 Gy in 4 frs	No
[40]	51 (treated) and 54 CTR	S + adj RT	50 Gy and 2 Gy	No
[41]	38	Intracavitary brachytherapy accelerated hypofractionated partial breast irradiation (IBAPBI)	34.0 Gy in 10 frs and 3.4 Gy twice per day (with a minimum of 6 hours between each treatment, for a total of 5 treatment days)	No

Abbreviations for Table 1: adjuvant (adj); breast cancer (BC); chemotherapy (CHT); combined chemo-radiotherapy (CRT); control (CTR); fractions (frs); Gray (Gy); human epidermal growth factor receptor-2 (Her-2); hormone therapy (HT); intracavitary brachytherapy accelerated hypofractionated partial breast irradiation (IBAPBI); intraoperative radiotherapy (IORT); radiotherapy (RT); surgery (S); stereotactic body radiation therapy (SBRT); total dose (TD). See Abbreviation list for other acronyms.

**Table 2 ijms-20-00382-t002:** Summary of results of the analyzed studies reporting details on sampling types and timepoints, type of cytokines, and main findings. See Table 1 for additional details.

Ref	Sample Type	Sampling Timepoints	Cytokines	Results
[27]	Blood	Before RT and on the last day of RT	TGF-β1 and PDGF-AB	TGF-β1 and PDGF levels decreased significantly during RT
[28]	Blood	From 6 and 18 months after treatment	IL-6, TGF-α, IL-1β	There were no significant baseline differences between survivors and the CTR group in LPS-stimulated TNF-α, IL-6, or IL-1b cytokines, or in the cytokine *z* score. There were significant differences in the trajectories of stimulated cytokines over time by treatment group with survivors treated with a combination of surgery, radiation, and chemotherapy having the highest increases in stimulated cytokines
[29]	Blood and serum	Before and after RT (24 h, 1 and 4 months after RT)	IL-1β, IL-6, IL-8, IL-10, and TNF-α	Compared to controls: increased IL-6; lower IL-8 at baseline increases during SBRT reaching level similar to those of controls; no differences, IL-10, IL-1β, and TNF-α not detectable
[30]	Specimen	During resection	Intratumoral IFN-γ mRNA and protein levels	No association with metastasis
[31]	Serum	Before and 1 month after treatment	IL-6, IL-8, IL-10, SCC-Ag, and CYFRA 21-1	Potential markers in the metastasis and BC prognosis
[32]	Specimen	During resection	CCL2, CCL3, CCL4, CCL5, CCL8, CCL18, CCL19, CCL21, CXCL9, CXCL10, CXCL11 and CXCL13	Radiosensitivity and immune activation
[33]	Biopsy	Pre-treatment only	RANK/RANK ligand RANK/OPG axis	RANK is increased in Her-2 negative and basal BC, and correlates with worse recurrence, free survival, and risk of bone metastases
[34]	Surnatant from S wound fluid	24 h after IORT	TGF-β1	Invariant after IORT
[35]	Serum	Before, at the end (week 7) and 6 weeks after RT	IL-6, IL-1ra1RA	In the CTR group, IL-6 increased at the end of RT and decreased 6 weeks after RT. IL-6 was invariant in patients performing exercise. IL-1ra was similar in both groups and increased only slightly after RT
[36]	Blood	From 8 am to 12 noon at 4-hour intervals before any treatment	IL-10RB, IFNA2, CXCL13, IL-17C, IL-17F, IL-13, CCL26, CSF2, IL-3, OSM, IL-1A, IL-16, IL-5RA, TNFSF13	All but three genes were downregulated in the blood of triple-negative BC patients. The most downregulated cytokines were IL-17C and IL-17F, IL-17C, better known as IL-21
[37]	Skin biopsy	From 0, 7, to 21 years after RT.	TGF-β1	Alterations in blood and lymphatic vessel are correlated with changes in TGF-β1 and endoglin levels, and with macrophage infiltration. Bisphosphonate treatment impaired leucocyte influx, but also negatively affected neovessel formation
[38]	Serum	From 8.3 up to 12 years (mean 9.9 years)	IL-6 and CTGF	No correlation between IL-6 and age. No correlation between cytokines and RT or fibrosis
[39]	Plasma	Baseline, weekly, 3 and 6 months post-treatment	IL-1β, IL-2, IL-6, IL-8, TNF-α, MCP-1, IL-10, VEGF, EGF, INF-β	IL-1b, Il-2, IL-6, and TNFα were increased 4 weeks after RT.
[40]	Plasma	1 year after RT	IL-6	Elevation of plasma IL-6 levels in patients 1 year post-RT
[41]	Serum	Serum was drawn before S before, 1 month after RT, and every subsequent 6 months for 2 years	TGF-β1	Elevated TGF-β1 levels in patients with moderate to severe radiation-induced fibrosis (RIF) compared with those who experienced none to mild RIF. This elevation was transiently eliminated after surgery and before IBAPBI, but once again it became significant during IBAPBI and persisted at 6 months, 12 months, 18 months, and 24 months

Main abbreviations for Table 2: breast cancer (BC); chemokine ligands (CCL); connective tissue growth factor (CTGF); healthy control (CTR); chemokine (C-X-C motif) ligand (CXC); intracavitary brachytherapy accelerated partial breast irradiation (IBAPBI); interleukin (IL); interferon (INF); intraoperative radiotherapy (IORT), lipopolysaccharide (LPS); monocyte chemoattractant protein-1 (MCP-1); platelet-derived growth factor anti-body (PDGF-AB); radiation-induced fibrosis (RIF); radiotherapy (RT); surgery (S); transforming growth factor (TGF); tumor necrosis factor (TNF). See Abbreviation list for other acronyms.

**Table 3 ijms-20-00382-t003:** Two studies which did not include RT but can be indicative for basal cytokine values.

Ref	No. of BC Patients and CTR	Treatment (S, RT, CHT, CRT, Combined)	RT TD and d in frs	CHT/HT	Sample Type	Sampling Timepoints	Cytokines	Results
[42]	98	Anti-Her-2 therapy (trastuzumab)	None	No	Biopsy	During resection	FOXP3	Associated with OS
[43]	96 BC + 30 healthy donors as CTR	Standard treatments after sampling	None	No	Serum	Before any treatment	TGF-β1	TGF-β1 levels statistically significant and higher in BC patients than in CTR

Abbreviations for Table 3: breast cancer (BC); healthy control (CTR); forkhead box P3 (FOXP3); human epidermal growth factor receptor-2 (Her-2); overall survival (OS); transforming growth factor beta-1 (TGF-β1).

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
