# Peer review of "Cytokine Modulation in Breast Cancer Patients Undergoing Radiotherapy: A Revision of the Most Recent Studies"

_ijms, 2019, doi:10.3390/ijms20020382_

Round 1
Reviewer 1 Report
In this review, the authors summarize recent literature related to modulation of cytokine expression in patients with breast cancer receiving radiotherapy. The authors reason that since antitumor responses induced by radiation therapy (RT) might be mediated by immune effects in addition to the well-known DNA-related changes, changes in cytokine profiles following RT, could have diagnostic and/or prognostic value. The review is based on 15 manuscripts encompassing a wide variety of cytokines and clinical variables, including other treatments (e.g., surgery, chemotherapy, hormone-therapy). Thus, it is not easy to compare results among the studies and reach strong, encompassing conclusions. However, the authors make the point that changes in IL-2, IL-6 and TGFb1 might be the most significant. Particularly, increases in IL-2 and IL-6 following RT and greater levels of IL-6 predicting worse outcome/toxicity.
Given the complexity of variability in terms of patient populations, cancer stages, treatments, the authors have done a good job in trying to come up with those cytokines that are the most informative. Thus, the conclusions are based only in a few references (not all 15 manuscripts studied the same cytokines). One weakness of the review is that the authors limit themselves to report the changes in cytokine patterns without addressing the potential causes, their clinical significance and potential confounding factors (e.g., inflammation related to other causes, including other treatments, stage of cancer, etc.), which could have added to the strength of this review. While the review is relatively well-written, it will need revision for errors in English grammar, spelling and other typos. The tables are meant to provide key information, but are a little cumbersome, particularly because of abbreviations that had not been previously introduced. It is thus recommended that each table list all of the abbreviations included. Some references are not listed by their number (in the reference list), but by the PMID, which should be corrected. The title should read “Cytokine modulation ……” and not Cytokines modulation …
Author Response
We thank the reviewr for his/her precious comments and suggestions to improve our manuscript.
We have corrected the manuscript by improving English, Table 2 is completed (the column of “sample type” was erroneously missing, now it has been included), and the references have been correctly reported according journal’s format. We highlighted text corrections in yellow instead of using track changes in order to facilitate manuscript reading.
The title has been opportunely modified as suggested.
An indication of abbreviations has been included below each Table, and references have been correctly indicated.
Reviewer 2 Report
The authors have very nicely summarized the last five-year works on cytokine modulation in breast cancer patients undergoing radiotherapy. Authors have used the right parameters to retrieved relevant references from PubMed. Authors have followed the right protocol to analyzed retrieved data.
I have following suggestion for the author.
1) Breast cancer (BC) is the most common tumor>>This statement may or may not be right. Is the author referencing to a particular country or worldwide (https://www.who.int/news-room/fact-sheets/detail/cancer). A clear mention is required in this statement.
2) Referencing is not done as per the journal's format. Sometimes the reference is given by writing numbers, sometimes with author names and year [Wang H & Yang X 2017], while sometimes as PMID [PMID: 24800238] Authors should use uniform referencing as per the journal’s guidelines.
3) The quality of the article can be enhanced substantially by improving the English. Authors can take help of an native speakers. Some errors are listed below.
a. authors indicate also> authors also indicate
b. were found also in>were also found in
c. we considered also works reporting> we also considered works
d. also cancer therapies have an influence> cancer therapies also have an influence
e. findings are reported also in>findings are also reported in
f. is found also for the anti-inflammatory>is also found forg.
4) 5) Sentences need improving.
a. [De Sanctis PMID: 24800238], on the contrary is has been showed invariant in Wang et al. [29].
b. No mention was done to RT treatment.
c. are until now contradictory.
d. Inhibitors of IL-6 trans-signaling and only binds to the complex of
6) 2018, PMID:).>referencing error
7) PMD: 27589056].> >referencing error
8) costly and easier to perform>>costly or cheap?
Briefly, English editing is required to further consider this manuscript.
Author Response
We thank the reviewer for his/her precious comments and suggestions to improve our manuscript.
We have corrected the manuscript by improving English, Table 2 is completed (the column of “sample type” was erroneously missing, now it has been included), and the references have been correctly reported according journal’s format. We highlighted text corrections in yellow instead of using track changes in order to facilitate manuscript reading.
English has been improved and grammar errors or typos corrected.
In the new manuscript version we have specified in the discussion those studies reporting survival curves showing correlation to individual cytokine expression and patient survival, and specific references have been included.
Reviewer 3 Report
Marconi et.al. provides an extensive review on the modulation of cytokines in breast cancer patients undergoing radiotherapy. The review article provides a good coverage of the last 5 yrs’ literature and also refers back to some of classical publications which significantly improves the manuscript. This review will be a good addition to the current field. A few minor issues need to be addressed:
1. Although a good manuscript, the authors would benefit from including a proof-reader. The English is sometimes a little hard to follow in some instances.
2. The authors have stated that various cytokines have a high likelihood of serving as prognostic markers in BC. The manuscript would really improve if the authors were to include some survival curves showing correlation of individual cytokine expression and patient survival. This would further consolidate the authors’ conclusions, especially the comments on interleukins.
Overall, a good review and will be of wide interest. I recommend accepting with minor modifications.
Author Response
We thank the reviewer for his/her precious comments and suggestions to improve our manuscript.
We have corrected the manuscript by improving English, Table 2 is completed (the column of “sample type” was erroneously missing, now it has been included), and the references have been correctly reported according journal’s format. We highlighted text corrections in yellow instead of using track changes in order to facilitate manuscript reading.
1. The statement regarding breast cancer incidence has been specified.
2. References have been correctly indicated according journal’s style and format.
3. English has been improved and grammar errors corrected.